# Continuous Relaxation for the Multivariate Noncentral Hypergeometric Distribution

**Thomas M. Sutter, Laura Manduchi, Alain Ryser & Julia E. Vogt**
Department of Computer Science
ETH Zurich
`{thomas.sutter,laura.manduchi,alain.ryser,julia.vogt}@inf.ethz.ch`

## Abstract

Partitioning a set of elements into a given number of groups of a priori unknown sizes is an essential task in many applications. Due to hard constraints, it is a non-differentiable problem that prohibits its direct use in modern machine learning frameworks. Hence, previous works mostly fall back on suboptimal heuristics or simplified assumptions. The multivariate hypergeometric distribution offers a probabilistic formulation of sampling a given number of elements from multiple groups. Unfortunately, as a discrete probability distribution, it neither is differentiable. We propose a continuous relaxation for the multivariate non-central hypergeometric distribution. We introduce an efficient and numerically stable sampling procedure that enables reparameterized gradients for the hypergeometric distribution and its integration into automatic differentiation frameworks. We additionally highlight its advantages on a weakly-supervised learning task.

## 1 Introduction

A variety of machine learning approaches, such as deep generative models, rely on differentiable sampling procedures, most notably the reparameterization trick for Gaussian distributions (Kingma & Welling, 2014; Rezende et al., 2014). As part of either generative models with discrete latent variables (Kingma et al., 2014) or attention mechanisms (Xu et al., 2015), the categorical distribution is another example. The concrete distribution (Maddison et al., 2017) or Gumbel-Softmax trick (Jang et al., 2016) has recently boosted its usage in stochastic networks. Unlike the high-variance gradients of score-based methods such as REINFORCE (Williams, 1992), these works enable reparameterized and low-variance gradients with respect to the categorical weights. Following the Gumbel-Softmax trick, reparameterization procedures combined with continuous relaxations for discrete distributions and other non-differentiable algorithms gained attraction in machine learning research (Grover et al., 2019; Xie & Ermon, 2019; Paulus et al., 2020). Despite enormous progress in recent years, the extension to more complex probability distributions is still missing or comes with a trade-off regarding differentiability or computational speed (Huijben et al., 2021).

The hypergeometric distribution is a discrete probability distribution that describes the probability of $x$ successes out of $n$ trials from a total population of size $N$ with $m$ members belonging to the success class (Upton & Cook, 2014). The most common application is sampling without replacement, for which an urn model is the standard example. In the univariate case, the urn consists of marbles in two different colours, whereas in the multivariate case, there are marbles in $c$ different colours. The hypergeometric distribution plays an essential role in various applications ranging from biology and gene mutations to computer science and recommender systems or social sciences and the analysis of networks (Becchetti et al., 2011; Lodato et al., 2015; Casiraghi et al., 2016). It is essential wherever the choice of a single element influences the distribution over classes of the remaining elements. Previous work uses the hypergeometric distribution implicitly to model assumptions or as a tool to prove theorems. However - to the best of our knowledge - it is not yet integrated into the learning processes themselves.

In this work, we propose a continuous relaxation for the multivariate non-central hypergeometric distribution that enables reparameterized gradients with respect to the distribution parameters. A differentiable formulation and reparameterizable gradients enable the integration of the hypergeo-

metric distribution into stochastic networks and modern learning frameworks. The hypergeometric distribution introduces dependencies between elements or samples where previous works had to assume independence.

We demonstrate the potential of the hypergeometric distribution for a weakly-supervised learning task where we assign latent dimensions to groups of unknown size. We perform a second experiment described in the appendix, where we highlight the advantage of the proposed model in a clustering algorithm. We model the number of samples per cluster using an adaptive hypergeometric distribution prior, overcome the simplified i.i.d. assumption of previous work, and establish a dependency structure between dataset samples.

## 2 METHOD

**Definition 2.1** (Multivariate Fisher's Non-Central Hypergeometric Distribution). A random vector $\boldsymbol{X}$ follows Fisher's non-central multivariate distribution, if its joint probability mass function is given by (Fisher, 1935)

$$P(\boldsymbol{X} = \boldsymbol{x}) = \frac{1}{P_0} \prod_{i=1}^{c} \binom{m_i}{x_i} \omega_i^{x_i} \tag{1}$$

where

$$P_0 = \sum_{(y_1,\ldots,y_c)\in\mathcal{S}} \prod_{i=1}^{c} \binom{m_i}{y_i} \omega_i^{y_i} \tag{2}$$

where $\binom{n}{k} = \frac{n!}{k!(n-k)!}$, $c \in \mathbb{N}_+$ is the number of different classes (e.g. marble colours in the urn), $\boldsymbol{m} = [m_1, \ldots, m_c] \in \mathbb{N}^c$ describes the number of elements per class (e.g. marbles per colour), $N = \sum_{i=1}^{c} m_i$ is the total number of elements (e.g. all marbles in the urn) and $n \in \{0, \ldots, N\}$ is the number of elements (e.g. marbles) to draw. $\boldsymbol{\omega} \in \mathbb{R}_{0+}^c$ describe the class weights, which introduce selection bias between classes. The support of the non-central hypergeometric distribution is equal to the support $\mathcal{S}$ of the central distribution (Equation (10)).

For general information on the hypergeometric distribution, we refer the interested reader to Appendix A.

### 2.1 SAMPLING FROM THE NON-CENTRAL HYPERGEOMETRIC DISTRIBUTION

There exist different sampling procedures for the non-central hypergeometric distribution in the literature. Each of these comes with its own advantages and disadvantages (for more details see (Fog, 2008b)). For the purpose of finding an efficient and numerically stable sampling procedure that is differentiable and batch-wise computable, we choose the conditional sampling procedure. The first class $X_1$ is sampled from the marginal distribution, then the following classes $X_i$ with $i > 1$ are sampled from the conditional distribution given all previously sampled classes $\{X_1 = x_1, \ldots, X_{i-1} = x_{i-1}\}$ (Johnson, 1987). As all classes are sampled sequentially, this algorithm scales linearly with the number of classes, and not with the number of possible outcomes of a random draw as in previous works (Liao & Rosen, 2001). Broadly speaking, the sampling algorithm consists of 3 parts:

1. Reformulate the multivariate distribution as a sequence of interdependent and conditional univariate distributions.

2. Calculate the PMF of the respective univariate distributions.

3. Sample from this conditional distribution utilizing the Gumbel-Softmax trick.

The sampling procedure is described in Algorithm 1 using pseudo-code, while the details of the individual steps are explained and discussed in the following sections of this work.

## 2.2 SEQUENTIAL SAMPLING USING CONDITIONAL DISTRIBUTIONS

As stated above, we reformulate the problem using a conditional sampling scheme by sequentially sampling from a conditional univariate distribution instead of a multivariate one. In the first step, class $i = 1$ is seen as the success class. The remaining classes $i > 1$ are merged together into a single, negative class. We sample $x_1$ elements from class $i = 1$ out of $n$ elements where $n$ is the total number of sampled elements. In the second step, class $i = 2$ is seen as success class and, again, the remaining classes $i > 2$ are seen as the negative class. We sample $x_2$ elements from class $i = 2$. Because we already sampled $x_1$ elements in the first step, the number of elements to sample are $n - x_1$ elements at this stage. From step 1 to step 2, the number of elements to sample from (e.g. the total number of marbles in the urn) changes from $N_1 = \sum_{i=1}^{c} m_i$ to $N_2 = N_1 - m_1$. We continue with this procedure until no class is left to be sampled from, i.e. $i = c$ (Fog, 2008b).

More formally, we define two classes as $L$ and $R$ with the respective parameters that are given by (Fog, 2008b)

$$m_L = \sum_{l \in L} m_l, \qquad m_R = \sum_{r \in R} m_r, \qquad \omega_L = \frac{\sum_{l \in L} \omega_l \cdot m_l}{m_L}, \qquad \omega_R = \frac{\sum_{r \in R} \omega_r \cdot m_r}{m_R} \qquad (3)$$

where $\omega_l$ and $\omega_r$ are the respective class weights. In the first step, we select a single class $i$ as set $L$, and the remaining classes as set $R$. Then, for each remaining steps, we select a single class $j$ from set $R$ as our new set $L$. Although we follow a simple approach, there exist various algorithms for creating univariate distributions out of subsets of classes. We leave the exploration of different and more sophisticated subset selection strategies for future work. For non-central distributions, this method produces biased samples as merging multiple classes into one is only approximately equal to the univariate non-central distribution. The selection of which classes are merged together might influence the quality of the approximation (Fog, 2008b).

## 2.3 CONTINUOUS RELAXATION

After simplifying the problem of sampling from the joint distribution into sequentially sampling from conditional distributions, we present continuous relaxations for these. Continuous relaxations describe procedures to make discrete distributions differentiable with respect to their parameters (Huijben et al., 2021).

**Lemma 2.2.** *The Gumbel-Softmax trick can be applied to the conditional distribution $P(X_i = x_i | \{X_k = x_k\}_{k=1}^{i-1})$ of class $i$ given the already sampled classes $k < i$.*

The proof for Lemma 2.2 can be found in Appendix B.2. The Gumbel-Softmax trick enables the reparameterization of categorical distributions and their gradients with respect to its parameters. We make use of the Gumbel-Softmax trick to reparameterize the conditional distributions, which results in an efficient and reliable sampling procedure. The reparameterization scheme for the univariate non-central hypergeometric distribution goes as follows:

$$\boldsymbol{u} \sim \boldsymbol{U}(0, 1), \qquad \boldsymbol{g}_i = -\log(-\log(\boldsymbol{u})), \qquad \hat{\boldsymbol{r}}_i = \boldsymbol{\alpha}_i + \boldsymbol{g}_i \qquad (4)$$

where $\boldsymbol{u} \in [0, 1]^{m_i + 1}$ is a random sample from an i.i.d. uniform distribution $\boldsymbol{U}$. As a result, $\boldsymbol{g}_i$ is i.i.d. gumbel noise. $\hat{\boldsymbol{r}}_i$ is the perturbed conditional probability for class $i$ given the class conditional unnormalized log-weights $\boldsymbol{\alpha}_i$. We use the softmax function to generate $(m_i + 1)$-dimensional sample vectors from the perturbed unnormalized weights $\hat{\boldsymbol{r}}_i / \tau$, where $\tau$ is the temperature parameter. Note that softmax is the continuous and differentiable approximation to the argmax function. We refer the reader to Jang et al. (2016) or Maddison et al. (2017) for more details on the Gumbel-Softmax trick itself. This procedure relates to the `contRelaxSample` function in Algorithm 1.

For the multivariate non-central hypergeometric distribution, the learnable parameters of the distribution are the relative class weights $\boldsymbol{\omega}$. It is important to mention that the assignment of $\boldsymbol{\omega}$ is independent of the sampling procedure and therefore $\boldsymbol{\omega}$ are not affected by the sequential sampling of classes. This is essential for the reparameterization of distributions (for more details see Huijben et al. (2021)). The remaining parameters of the multivariate distribution are fixed by design of experiments, $\boldsymbol{m}$, or do not influence the ordering of elements, $n$.

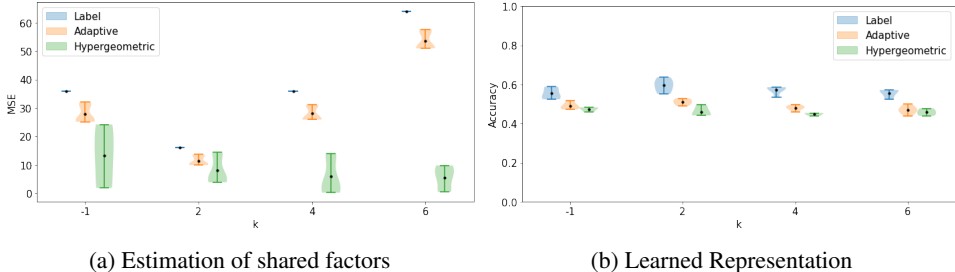

(a) Estimation of shared factors          (b) Learned Representation

Figure 1: Evaluation of three different methods on the weakly-supervised learning experiment with respect to two different tasks. We compare our HypergeometricVAE against two recent methods, LabelVAE and AdaptiveVAE. First (Figure 1a), we evaluate the models' ability to estimate the number of shared generative factors between a pair of images. The performance is reported using mean squared error (MSE) between the true and the estimated number of shared factors (lower is better). Second (Figure 1b), we assess the quality of the learned latent representation. The performance is reported using classification accuracy with respect to the generative factors. We report the mean accuracy over all generative factors (higher is better). The proposed model (Hypergeometric) outperforms existing models (Label, Adaptive) by a margin in estimating the number of shared factors while reaching almost the same performance in the quality of latent representations.

## 2.4 CALCULATE PROBABILITY MASS FUNCTION

In Sections 2.2 and 2.3, we derive a sequential sampling procedure. In this sequential procedure we repeatedly simulate a univariate distribution to simplify sampling. As a result, the PMF of the joint multivariate distribution is not needed in the sampling procedure. Instead we can restrict the computation of the PMF to the univariate case. For the multivariate extension, see Appendix B.1. The PMF $P_{X_L}(x)$ for the hypergeometric distribution of two classes $L$ and $R$ defined by $m_L, m_R, \omega_L, \omega_R$ and $n$ is given as

$$P_{X_L}(x) = \frac{1}{P_0} \binom{m_L}{x} \omega_L^x \binom{m_R}{n-x} \omega_R^{n-x} \tag{5}$$

$P_0$ is defined as in Equation (2).

**Lemma 2.3.** *The unnormalized log-probabilities*

$$\log P_{X_L} \propto x \cdot \log \omega_L + (n-x) \log \omega_R + \psi_F(x) \tag{6}$$

*define the unnormalized weights of a categorical distribution that follows Fisher's non-central hypergeometric distribution.* $\psi_F(x)$ *is defined as follows*

$$\psi_F(x) = -\log\Gamma(x+1) - \log\Gamma(m_L - x + 1) - \log\Gamma(n - x + 1) - \log\Gamma(m_R - n + x + 1) \tag{7}$$

The proof for Lemma 2.3 can be found in Appendix B.3. Additionally, calculations in the log-domain increase numerical stability and also preserve the ordering between different values $x$ (Huijben et al., 2021). This subsection relates to the calcLogPMF function in Algorithm 1. Equation (7) and more general Equation (6) can be calculated efficiently and reliably as common automatic differentiation frameworks have numerically stable implementations of $\log\Gamma(x)$, e.g. Tensorflow (Abadi et al., 2016) or PyTorch (Paszke et al., 2019). Additionally, all necessary steps can be calculated batch-wise.

Using the multivariate form of Lemma 2.3 (see Appendix B.1), it is possible to directly calculate the categorical weights for the multivariate states. A multivariate state is defined as valid combination of $\boldsymbol{X} = \boldsymbol{x}$. It would result in a significant speed-up for large number of classes $c$ compared to the conditional sampling procedure. However, the number of multivariate states is $\prod_{i=1}^{c} m_i$, which quickly results in unfeasible memory requirements. Therefore, we would be restricted to settings with no practical relevance.

## 3 EXPERIMENT ON WEAKLY-SUPERVISED LEARNING

In this experiment, we induce weak-supervision through pairs of images that may share part of their generating factors. In general, neither the true number of generative factors nor the number of

shared latent factors is known. We use the setting and code from Locatello et al. (2020) and compare the models on the mpi3d toy dataset (Gondal et al., 2019). All models are trained as a variational autoencoder (Kingma & Welling, 2014, VAE) to maximize an evidence lower bound (ELBO) on the marginal log-likelihood of the pair of images. We perform the same experiment for multiple dataset versions with different numbers of independent generative factors $k$. By independent we mean generative factors that are not shared between pairs of images.

We compare three methods that aim at inferring shared and independent latent factors. All methods aggregate shared factors that replace the individual factors. The difference among methods is only in the selection process of shared factors. The first one assumes that the number of independent factors is known (Bouchacourt et al., 2018; Hosoya, 2018, LabelVAE). Similar to the experiments in Locatello et al. (2020), we assume this to be 1 for all experiments. The second method relies on an adaptive heuristic to infer the number of shared latent factors (Locatello et al., 2020, AdaptiveVAE). First, they calculate the Kullback-Leibler (KL) divergence between pairs (across images) of latent factors. Then, their heuristic defines a threshold to determine whether a pair of factors encodes the same information, i.e. is shared, or not. In our approach (HypergeometricVAE), we model the number of shared and independent latent factors of a pair of images as a multivariate hypergeometric distribution with unknown $\omega \in \mathbb{R}^2_{0+}$. For $d$ being the number of latent factors, we have $m_i = d$ where $i \in \{1, 2\}$ and the number of elements to draw $n = d$. Given class weights $\omega$, we are able to sample estimates for the $k$ independent and $d - k$ shared factors. The proposed formulation allows to infer such $\omega$ and simultaneously learn the latent representation in a fully differentiable setting. Like in AdaptiveVAE, we utilize the KL divergences between latent factors. They are passed to a single dense layer which outputs $\omega$. We make use of the stochastic sorting algorithm (Grover et al., 2019) for sorting the latent factors by KL divergence. Next, we define the top-$k$ latent factors to be independent, and the remaining $d - k$ to be shared ones where $k$ and $d - k$ are samples from the hypergeometric distribution. For a more detailed description of HypegeometricVAE, the baseline models and the dataset, see Appendix C.2.

To assess the performance of the three methods, we evaluate the trained models on two different tasks and over four *mpi3d* based training sets with the true number of independent factors $k$ set to 2, 4, 6 and uniformly random between 1 and 7 ($k = -1$). We measure the mean squared error of predicting the number of shared latent factors (Figure 1a) and quantify downstream performance of the learned latent representations on a classification task (Figure 1b). As can be seen, previous methods are not able to accurately estimate the number of shared factors. In fact, both baseline methods estimate the number of shared factors to be approximately constant for all experiments - independent of the underlying ground truth number of shared factors. For the first model, this does not come as a surprise, whereas this was quite unexpected for the second approach given their in theory adaptive heuristic. On the other hand, the proposed approach is able to accurately estimate the number of shared factors which is reflected in the low mean squared error (MSE) for all experiments, including the most difficult experiment with random number of independent factors. To assess the quality of the learned latent representation, we evaluate it with respect to all ground truth generative factors of the dataset. We train classifiers for all factors and calculate their accuracy. The reported accuracy is the average over the factor-specific accuracies. Given the general nature of the method, the positive results of the proposed method are quite surprising. Different to previous works, it was not explicitly designed for weakly-supervised learning but is nevertheless able to achieve results that are more than comparable to field-specific models. Additionally, the proposed method is able to give accurate estimates on the latent space structure for different experimental settings.

## 4 CONCLUSION

We propose a continuous relaxation for the multivariate noncentral hypergeometric distribution in this work. In combination with the Gumbel-Softmax trick, this new formulation enables reparameterized gradients with respect to the class weights $\omega$ of the hypergeometric distribution. We show the potential of the hypergeometric distribution in machine learning by applying it to a weakly-supervised learning task. In this application, the method making use of the hypergeometric distribution reaches state-of-the-art performance. We believe this work is an essential step toward integrating the hypergeometric distribution into more machine learning algorithms. Applications in biology and social sciences represent potential directions for future work.

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

## A  PRELIMINARIES

The hypergeometric distribution is a discrete probability distribution that describes the probability of $x$ successes in $n$ draws without replacement from a finite population of size $N$ with $m$ elements that are part of the success class. This is different to the binomial distribution which describes the probability distribution of $x$ successes in $n$ draws with replacement.

**Definition A.1** (Hypergeometric Distribution (Gonin, 1936)[1])**.** A random variable $X$ follows the hypergeometric distribution, if its probability mass function (PMF) is given by

$$P_X(x) = P(X = x) = \frac{\binom{m}{x}\binom{N-m}{n-x}}{\binom{N}{n}} \tag{8}$$

Urn models are classical examples of the hypergeometric probability distribution. If we think of an urn with marbles in two different colours, e.g. green and purple, we can label as success the drawing of a green marble. Then $N$ defines the total number of marbles and $m$ the number of green marbles in the urn. $x$ is the number of green marbles and $n - x$ the number of purple marbles that are drawn.

The multivariate hypergeometric distribution describes an urn with more than two colours, e.g. green, purple and yellow in the simplest case with three colours. As described in Johnson (1987), the definition is given by:

**Definition A.2** (Multivariate Hypergeometric Distribution)**.** A random vector $\boldsymbol{X}$ follows the multivariate hypergeometric distribution, if its joint probability mass function is given by

$$P(\boldsymbol{X} = \boldsymbol{x}) = \frac{\prod_{i=1}^{c}\binom{m_i}{x_i}}{\binom{N}{n}} \tag{9}$$

where $c \in \mathbb{N}_+$ is the number of different classes (e.g. marble colours in the urn), $\boldsymbol{m} = [m_1, \ldots, m_c] \in \mathbb{N}^c$ describes the number of elements per class (e.g. marbles per colour), $N = \sum_{i=1}^{c} m_i$ is the total number of elements (e.g. all marbles in the urn) and $n \in \{0, \ldots, N\}$ is the number of elements (e.g. marbles) to draw. The support $\mathcal{S}$ of the PMF is given by

$$\mathcal{S} = \left\{ \boldsymbol{x} \in \mathbb{Z}_{0+}^c : \forall i \quad x_i \le m_i, \sum_{i=1}^{c} x_i = n \right\} \tag{10}$$

---

[1] Although the distribution itself is older, Gonin (1936) were the first to name it hypergeometric distribution

---

**Algorithm 1** Sampling From Multivariate Noncentral Hypergeometric Distribution. The different blocks are explained in more detail in Sections 2.2 to 2.4

---

    **Input:** $\boldsymbol{m} \in \mathbb{Z}_{0+}^c, \boldsymbol{\omega} \in \mathbb{R}_{0+}^c, n \in \mathbb{N}, \tau \in \mathbb{R}_+$
    **Output:** $\boldsymbol{x} \in \mathbb{Z}_{0+}^c, \{\boldsymbol{\alpha}_i \in \mathbb{R}^{m_i}\}_{i=1}^c, \{\hat{\boldsymbol{r}}_i \in \mathbb{R}^{m_i}\}_{i=1}^c$
    **for** $i \in \{1, \ldots, c\}$ **do**
        $L \leftarrow i, R \leftarrow \{\bigcup_{j=i+1}^c j\}$                        # Formulate the multivariate as a
        $\boldsymbol{m} \to m_L, m_R \in \mathbb{Z}_{0+}, \boldsymbol{\omega} \to \omega_L, \omega_R \in \mathbb{R}_{0+}$      # univariate distribution (Section 2.2)
        $x_L, \boldsymbol{\alpha}_L, \hat{\boldsymbol{r}}_L \leftarrow \text{sampleUNCHG}(m_L, m_R, \omega_L, \omega_R, n, \tau)$ # Sample from univariate distribution
        $n \leftarrow n - x_L, \boldsymbol{m} \leftarrow \boldsymbol{m} \setminus \boldsymbol{m}_L, \boldsymbol{\omega} \leftarrow \boldsymbol{\omega} \setminus \boldsymbol{\omega}_L$          # re-assign classes for next step
        $x_i \leftarrow x_L, \boldsymbol{\alpha}_i \leftarrow \boldsymbol{\alpha}_L, \hat{\boldsymbol{r}}_i \leftarrow \hat{\boldsymbol{r}}_L$                  # assign values for class $i$
    **end for**
    **return** $\boldsymbol{x}, \{\boldsymbol{\alpha}_i\}_{i=1}^c, \{\hat{\boldsymbol{r}}_i\}_{i=1}^c$

    **function** SAMPLEUNCHG($m_i, m_j, \omega_i, \omega_j, n, \tau$)
        $\boldsymbol{\alpha}_i \leftarrow \text{calcLogPMF}(m_i, m_j, \omega_i, \omega_j, n)$                    # Section 2.4
        $x_i, \hat{\boldsymbol{r}}_i \leftarrow \text{contRelaxSample}(\boldsymbol{\alpha}_i, \tau))$                  # Section 2.3
        **return** $x_i, \boldsymbol{\alpha}_i, \hat{\boldsymbol{r}}_i$
    **end function**

---

Definitions A.1 and A.2 assume that the only property to distinguish between marbles is their colour. In other words, there is no property that leads to a selection bias between the different classes. Every marble is to be picked equally likely and the number of selected class samples is proportional to the ratio between class elements and total number of elements in the urn. Often this assumption is too restrictive and selection bias is a desired model property. Biased generalizations, which make certain colours more likely to be picked, exist for the uni- and the multivariate case. We focus on the multivariate distribution as the univariate distribution is a special case thereof.

In the literature, two different versions of the *noncentral* hypergeometric distribution exist, Fisher's (Fisher, 1935) and Wallenius' (Wallenius, 1963; Chesson, 1976) distribution. Due to limitations of the latter (Fog, 2008a), we will refer to Fisher's version of the noncentral hypergeometric distribution in the remaining part of this work.

# B   METHODS

## B.1   PMF FOR THE MULTIVARIATE FISHER'S NONCENTRAL DISTRIBUTION

In this section, we give a detailed derivation for the calculation of the log-probabilities of the multivariate Fisher's noncentral hypergeometric distribution. We end up with a formulation that is proportional to the actual log-probabilities. Because the ordering of categories is not influenced by scaling with a constant factor (addition/subtraction in log domain), these are unnormalized log-probabilities of the multivariate Fisher's noncentral hypergeometric distribution.

$$P(\boldsymbol{X} = \boldsymbol{x}) = \frac{1}{P_0} \prod_{i=1}^c \binom{m_i}{x_i} \omega_i^{x_i} \tag{11}$$

where $P_0$ is defined as in Equation (2). From there it follows

$$\log P(\boldsymbol{X} = \boldsymbol{x}) = \log\left(\frac{1}{P_0}\prod_{i=1}^{c}\binom{m_i}{x_i}\omega_i^{x_i}\right) \tag{12}$$

$$= \log\left(\frac{1}{P_0}\right) + \log\left(\prod_{i=1}^{c}\binom{m_i}{x_i}\omega_i^{x_i}\right) \tag{13}$$

$$= \log\left(\frac{1}{P_0}\right) + \sum_{i=1}^{c}\log\left(\binom{m_i}{x_i}\omega_i^{x_i}\right) \tag{14}$$

$$= \log\left(\frac{1}{P_0}\right) + \sum_{i=1}^{c}\left(\log\binom{m_i}{x_i} + \log\left(\omega_i^{x_i}\right)\right) \tag{15}$$

$$= \log\left(\frac{1}{P_0}\right) + \sum_{i=1}^{c}\left(\log\binom{m_i}{x_i} + x_i\log\left(\omega_i\right)\right) \tag{16}$$

Constants factor can be removed as the argmax is invariant to scaling with a constant factor which equals addition or subtraction in log-space. It follows

$$\log P(\boldsymbol{X} = \boldsymbol{x}) \propto \sum_{i=1}^{c}\left(\log\binom{m_i}{x_i} + x_i\log\left(\omega_i\right)\right) \tag{17}$$

$$\propto \sum_{i=1}^{c}\left(\log\frac{1}{x_i!(m_i - x_i)!} + x_i\log\left(\omega_i\right)\right) \tag{18}$$

$$= \sum_{i=1}^{c}\left(-\log\left(\Gamma(x_i + 1)\Gamma(m_i - x_i + 1)\right) + x_i\log\left(\omega_i\right)\right) \tag{19}$$

$$\tag{20}$$

In the last line we used the relation $\Gamma(k + 1) = k!$.

$$\log P(\boldsymbol{X} = \boldsymbol{x}) \propto \sum_{i=1}^{c} x_i\log\omega_i + \psi_F(\boldsymbol{x}) \tag{21}$$

where $\psi_F(\boldsymbol{x}) = -\sum_{i=1}^{c}\log\left(\Gamma(x_i + 1)\Gamma(m_i - x_i + 1)\right)$.

The Gamma function is defined as (Whittaker & Watson, 1996)

$$\Gamma(z) = \int_0^{\infty} x^{z-1}e^{-x}dx \tag{22}$$

### B.2 PROOF FOR LEMMA 2.2

*Proof.* When sampling class $i$, we draw $x_i$ samples from class $i$ where $x_i \leq m_i$. The conditional distribution $P(X_i = x_i|\{X_k = x_k\}_{k=1}^{i-1})$ for class $i$ given the already sampled classes $k < i$ simultaneously defines the weights of a categorical distribution. Sampling $x_i$ elements from class $i$ can be seen as selecting the $x_i$th category from the distribution defined by the weights $P(X_i = x_i|\{X_k = x_k\}_{k=1}^{i-1})$. Therefore, $\sum_{x_i \leq m_i} P_{X_i}(x_i) = 1$, which allows us to apply the Gumbel-Max trick and, respectively the Gumbel-Softmax trick. $\qquad\square$

### B.3 PROOF FOR LEMMA 2.3

*Proof.* Factors that are constant for all $x$ do not change the relative ordering between different values of $x$. Hence, removing them preserves the ordering of values $x$ (Barrett, 2017).

$$\log P_{X_L}(x) = \log\left(\frac{1}{P_0}\binom{m_L}{x}\omega_L^x\binom{m_R}{n - x}\omega_R^{n-x}\right) \tag{23}$$

$$\propto \log\binom{m_L}{x} + \log\binom{m_R}{n - x} + \log\left(\omega_L^x\right) + \log\left(\omega_R^{n-x}\right)$$

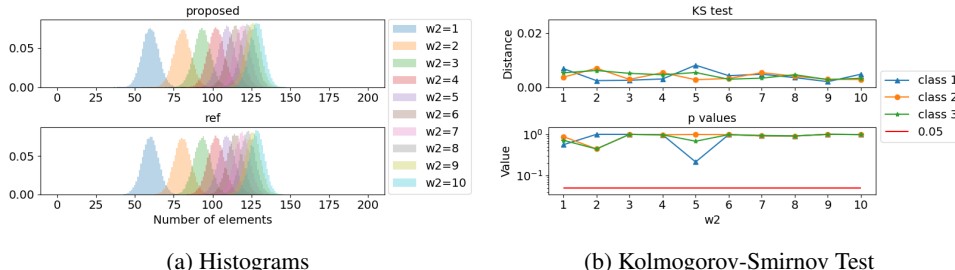

(a) Histograms          (b) Kolmogorov-Smirnov Test

Figure 2: Comparing random variables from the proposed distribution to a reference distribution. We draw samples from a multivariate noncentral hypergeometric distribution consisting of 3 classes. $m_i = 200 \; \forall i$ and $n = 180$. The class weights $\omega_1$ and $\omega_3$ for classes 1 and 3 are set to 1.0, $\omega_2$ is increased from 1.0 to 10.0 with a step size of 1.0 (defined as w2 in the figure legend). Figure 2a shows histograms of the number of elements for class 2. We see that the behaviour across different class weights $\omega_2$ is very similar between the proposed and the reference implementation. This behaviour is also reflected in values of the KS test and their respective $p$-values in Figure 2b.

Using the definition of the binomial coefficient (see Appendix A) and its relation to the Gamma function[2] $\Gamma(k+1) = k!$, it follows

$$
\begin{aligned}
\log P_{X_L} \propto &\; x \cdot \log \omega_L + (n - x) \cdot \log \omega_R \\
&- \log \Gamma(x+1) - \log \Gamma(m_L - x + 1) - \log \Gamma(n - x + 1) - \log \Gamma(m_R - n + x + 1)
\end{aligned}
\tag{24}
$$

With $\psi_F(x)$ as defined in Equation (7) it follows $\log P_{X_L} \propto x \cdot \log \omega_L + (n - x) \cdot \log \omega_R + \psi_F(x)$. $\quad\square$

## C  EXPERIMENTS

### C.1  KOLMOGOROV-SMIRNOV TEST

To assess the accuracy of the proposed method, we evaluate it against a reference distribution using the Kolmogorov-Smirnov test (Kolmogorov, 1933; Smirnov, 1939, KS). It is a nonparametric test to evaluate the equality of two distributions. It can be used for continuous as well as discrete 1-dimensional probability distributions. We compare the class conditional hypergeometric distributions against a reference distribution from SciPy (Virtanen et al., 2020). As described in Section 2 class conditional distributions are used to sample from the multivariate distribution. For this experiment, we use a multivariate hypergeometric distribution of three classes. We perform a sensitivity analysis with respect to the class weights $\boldsymbol{\omega}$. We keep $\omega_1$ and $\omega_3$ fixed at 1.0, $\omega_2$ is increased from 1.0 to 10.0 in steps of 1.0. For every value of $\omega_2$, we sample 50000 i.i.d. random vectors. The KS test quantifies a distance between the empirical distributions of two groups of samples. Hence, a small test value implies that the underlying distributions are similar. The null distribution of this test is calculated under the null hypothesis that the two groups of samples are drawn from the same distribution. In our setting, we would like the test to fail to reject the null hypothesis. In this case, the two groups of samples were generated by the same distribution, i.e. the two underlying distributions are equal. To reject the null hypothesis, the $p$-value should be below the significance threshold $t = 0.05$. And vice versa, $p > 0.05$ implies that the null hypothesis cannot be rejected which is desirable for our application.

Figure 2 shows the histogram of values of class 2 for all values of $\omega_2$ (Figure 2a) and the results of the KS test for all classes (Figure 2b). The histograms for class 1 and 3 can be found in the Appendix (Figure 3). We see that the histograms of the proposed and the reference distribution are visually similar. Additionally, the calculated distances of the KS-test are small and the corrected $p > 0.05$. We correct the $p$-values for false discovery rate of multiple comparisons as we are performing $c = 3$ tests per joint distribution. We use the Benjamini-Hochberg correction (Benjamini & Hochberg, 1995). In Figure 2b, we see that the test is clearly significant in 30 out 30 cases. For all tests $p$-values

---
[2]see Appendix B.1 for the definition on the Gamma function

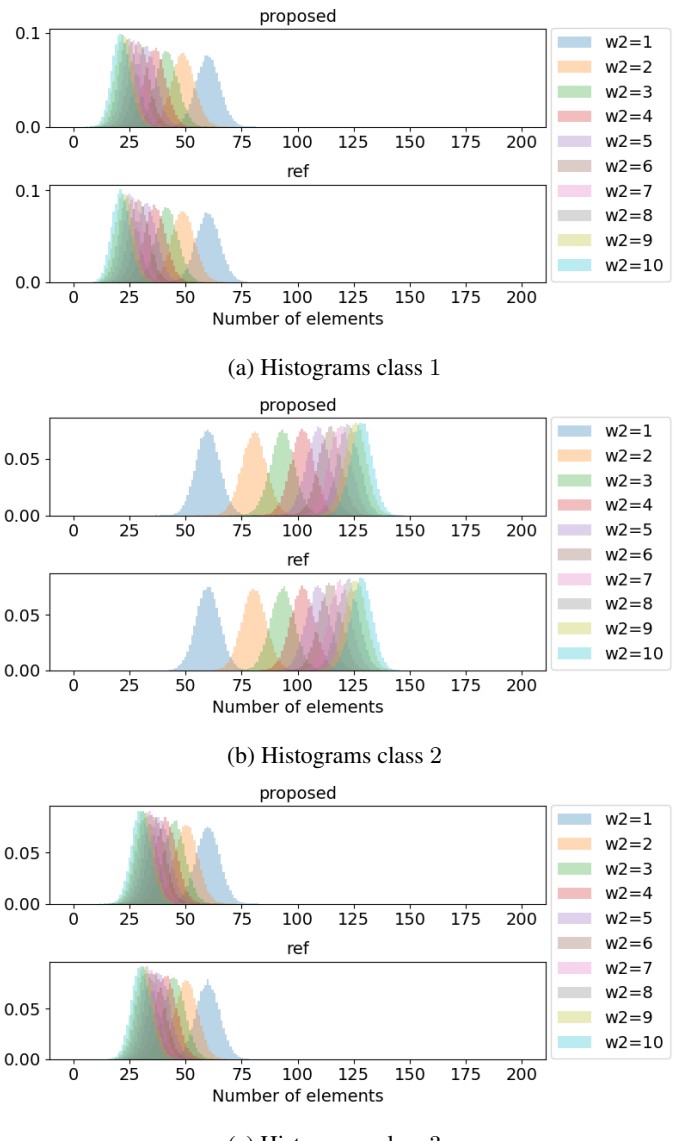

Figure 3: Comparing random variables drawn from the proposed distribution to a reference distribution.

are well above the threshold and many are close to 1.0. The results of the KS test give strong evidence that the proposed continuous relaxation and sampling procedure follow a noncentral hypergeometric distribution.

## C.2 WEAKLY-SUPERVISED LEARNING

### C.2.1 METHOD, IMPLEMENTATION AND HYPERPARAMETERS

In this section we give more details on the used methods. We make use of the `disentanglement_lib` (Locatello et al., 2019) which is also used in the original paper we compare to (Locatello et al., 2020). The baseline algorithms (Bouchacourt et al., 2018; Hosoya, 2018) are already implemented in `disentanglement_lib`. For details on the implementation of models, we refer to the original paper. We did not change any hyperparameters or network settings. All experiments were performed using $\beta = 1.0$ as this is the best performing $\beta$ according to Locatello et al. (2020). For all experiments we train 5 models with different random seeds.

All experiments are performed using GroupVAE (Hosoya, 2018). Using GroupVAE, shared latent factors are aggregated using an arithmetic mean. Bouchacourt et al. (2018) assume also knowledge about shared and independent latent factors. Different to GroupVAE, their ML-VAE aggregates shared latent factors by using the Product of Experts (i.e. geometric mean).

### C.2.2 HYPERGEOMETRICVAE (IN MORE DETAIL)

In our approach (HypergeometricVAE), we model the number of shared and independent latent factors of a pair of images as a hypergeometric distribution with unknown $\omega$. In reference to the urn model, shared and independent factors each correspond to one colour and the urn contains $d$ marbles of each colour, where $d$ is the dimensionality of the latent space. Given the correct weights $\omega$ and when drawing from the urn $d$ times, the number of each respective colour corresponds, in expectation, to the correct number of independent/shared factors. The proposed formulation allows to simultaneously infer such $\omega$ and learn the latent representation in a fully differentiable setting within the weakly-supervised pipeline by Locatello et al. (2019).

To integrate the procedure described above in this framework, we need two additional building blocks. First, we introduce a function that returns $\omega$. To achieve this, we use a single dense layer followed by a tangens hyperbolicus activation function which returns logits $\tilde{\omega}$. The input to this layer is a vector $\gamma$ containing the symmetric version of the KL divergences between pairs of latent distributions, i.e. for latent $P$ and $Q$, the vector contains $\frac{1}{2}(KL(P||Q) + KL(Q||P))$. An estimation $\hat{\omega}$ is than given by $\exp(\tilde{\omega})$. Second, sampling from the hypergeometric distribution with these weights leads to an estimate $\hat{k}$. Consequently, we need a method to select $\hat{k}$ factors out of the $d$ that are given. Similar to the original paper, we select the factors achieving the highest symmetric KL-divergence. In order to do this, we sort $\gamma$ in descending order using the stochastic sorting procedure `neuralsort` (Grover et al., 2019). This enables us to select the top $\hat{k}$ independent as well as the bottom $d - \hat{k}$ shared latent factors. Like AdaptiveVAE, we substitute the shared factors by the mean value of the original latent code before continuing the VAE forward pass in the usual fashion.

### C.2.3 DATA

The mpi3d dataset (Gondal et al., 2019) consists of frames displaying a robot arm and is based on 7 generative factors.

- object colour
- object shape
- object size
- camera height
- background colour
- horizontal axis
- vertical axis

Table 1: Evaluation of the clustering experiment on the MNIST datasets. We compare the methods on 3 different dataset versions, namely i) uniform class distribution ii) subsampling with 80% of samples and iii) subsampling with only 60% of samples. We subsample half of the classes. Accuracy (Acc), normalized mutual information (NMI), and adjusted rand index (ARI) are used as evaluation metrics. Higher is better for all metrics. Mean and standard deviations are computed across 5 runs.

| MODEL | ACC (%) | NMI (%) | ARI (%) |
|---|---|---|---|
| | UNIFORM | | |
| UNIFORM | $92.0_{\pm 3.0}$ | $84.8_{\pm 2.2}$ | $84.2_{\pm 4.3}$ |
| CATEGORICAL | $87.2_{\pm 5.0}$ | $81.8_{\pm 1.9}$ | $78.3_{\pm 4.6}$ |
| HYPERGEOMETRIC | $\mathbf{93.8}_{\pm 2.0}$ | $\mathbf{86.3}_{\pm 3.0}$ | $\mathbf{86.9}_{\pm 4.0}$ |
| | SUBSAMPLING (80 %) | | |
| UNIFORM | $90.8_{\pm 4.0}$ | $84.1_{\pm 2.2}$ | $83.2_{\pm 3.6}$ |
| CATEGORICAL | $87.4_{\pm 4.7}$ | $81.8_{\pm 2.3}$ | $78.2_{\pm 5.0}$ |
| HYPERGEOMETRIC | $\mathbf{92.5}_{\pm 0.5}$ | $\mathbf{84.6}_{\pm 0.8}$ | $\mathbf{84.4}_{\pm 1.0}$ |
| | SUBSAMPLING (60 %) | | |
| UNIFORM | $83.5_{\pm 3.9}$ | $80.7_{\pm 1.4}$ | $77.6_{\pm 2.6}$ |
| CATEGORICAL | $86.5_{\pm 4.9}$ | $81.3_{\pm 2.9}$ | $77.7_{\pm 6.3}$ |
| HYPERGEOMETRIC | $\mathbf{89.7}_{\pm 4.3}$ | $\mathbf{82.9}_{\pm 2.2}$ | $\mathbf{81.5}_{\pm 3.9}$ |

For more details on the dataset and in general, we refer to https://github.com/rr-learning/disentanglement_dataset.

### C.2.4 DOWNSTREAM TASK ON THE LEARNED LATENT REPRESENTATIONS

For the downstream task we sample randomly 10000 samples from the training set and 5000 samples from the test set. For every generative factor of the dataset, an individual classifier is trained on the 10000 training samples. Afterwards, every classifier evaluates the latent representations of the 5000 test samples. The classification accuracies of the individual classifiers is averaged into a single average accuracy. The average accuracy is the one reported.

### C.3 DEEP VARIATIONAL CLUSTERING

We investigate the use of the multivariate noncentral hypergeometric distribution in a deep clustering task. Several techniques have been proposed in the literature to combine long-established clustering algorithms, such as K-means or Gaussian Mixture Models, with the flexibility of deep neural networks to learn better representations of high-dimensional data (Min et al., 2018). Among those, Jiang et al. (2016), Dilokthanakul et al. (2016) and Manduchi et al. (2021) include a trainable Gaussian Mixture prior distribution in the latent space of a VAE. This permits a probabilistic approach to clustering where a clear generative assumption of the data is defined and optimised within the framework of stochastic gradient variational Bayes (Kingma & Welling, 2014; Rezende et al., 2014). A major drawback of the above models is that the samples are either assumed to be *i.i.d.* or they require pairwise side information, which limits their applicability in real-world scenarios.

The multivariate noncentral hypergeometric distribution can be easily integrated in VAE-based clustering algorithms to overcome limitations of current approaches. Given a dataset $X = \{\mathbf{x}_i\}_{i=1}^N$ that we wish to cluster into $K$ groups, we consider the following generative assumptions:

$$\mathbf{c} \sim p(\mathbf{c}; \boldsymbol{\pi}) \tag{25}$$

$$\mathbf{z}_i \sim p(\mathbf{z}_i|c_i) = \mathcal{N}(\mathbf{z}_i|\boldsymbol{\mu}_{c_i}, \boldsymbol{\sigma}_{c_i}^2 \mathbb{I}) \tag{26}$$

$$\mathbf{x}_i \sim p_\theta(\mathbf{x}_i|\mathbf{z}_i) = Ber(\boldsymbol{\mu}_{x_i}), \tag{27}$$

where $\mathbf{c} = \{c_i\}_{i=1}^N$ are the cluster assignments, $\mathbf{z}_i \in \mathbb{R}^D$ are the latent embeddings of a VAE and $\mathbf{x}_i$ is assumed to be binary for simplicity. In other words, we assume the data is generated from a random process where the cluster assignments are first drawn from a prior probability $p(\mathbf{c}; \boldsymbol{\pi})$, then each latent embedding $\mathbf{z}_i$ is sampled from a Gaussian distribution, whose mean and variance

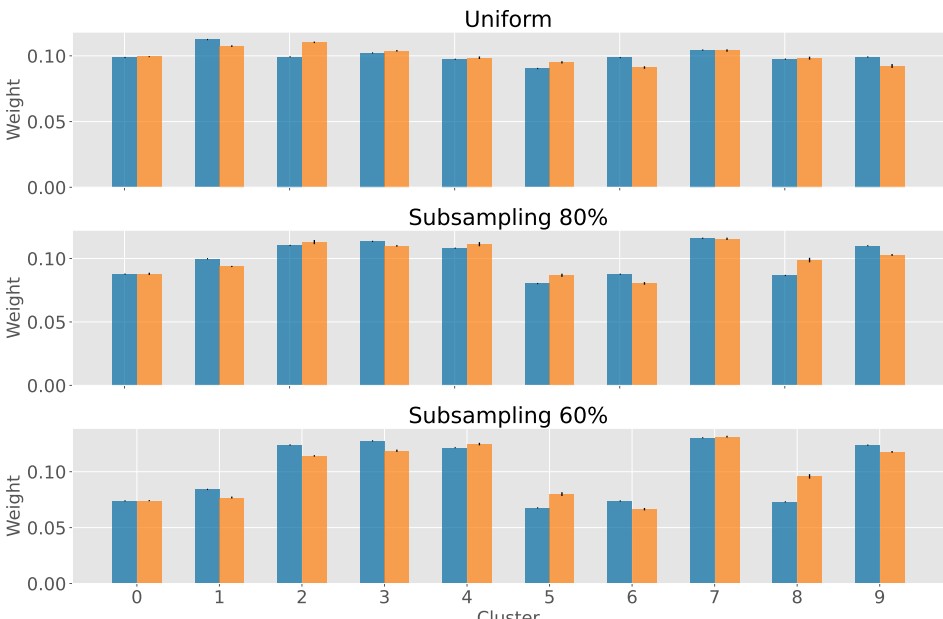

Figure 4: Evaluation of the cluster weights $\boldsymbol{\pi}$ of the multivariate noncentral hypergeometric prior distribution learned by the clustering method. The normalized size of each true class (■) is compared with the corresponding learned cluster weight $\pi_i$ (■). We use 3 different dataset versions: (upper) uniform class distribution; (mid) subsampling with 80% of samples and (lower) 60% of samples for half of the classes.

depend on the selected cluster $c_i$. Finally the sample $\mathbf{x}_i$ is generated from a Bernoulli distribution whose parameter $\boldsymbol{\mu}_{x_i}$ is the output of a neural network parameterized by $\boldsymbol{\theta}$, as in the classical VAE. With these assumptions, the latent embeddings $\mathbf{z}_i$ follow a mixture of Gaussian distributions, whose means and variances, $\{\boldsymbol{\mu}_i, \boldsymbol{\sigma}_i^2\}_{i=1}^K$, are tunable parameters. The above generative model can then be optimised by maximising the ELBO using the stochastic gradient variational Bayes estimator (we refer to Appendix C.3.1 for a complete description of the optimisation procedure).

Previous work (Jiang et al., 2016) modeled the prior distribution as $p(\mathbf{c}; \boldsymbol{\pi}) = \prod_i p(c_i) = \prod_i Cat(c_i|\boldsymbol{\pi})$ with either tunable or fixed parameters $\boldsymbol{\pi}$. In this task, we instead replace this prior with the multivariate noncentral hypergemetetric distribution with weights $\boldsymbol{\pi}$ and $K$ classes where every class relates to a cluster. Hence, we sample the number of samples per cluster (or cluster size) following Definition 2.1. The hypergeometric distribution permits to create a dependence between samples. The prior probability of a sample to be assigned to a given cluster is not independent of the remaining samples anymore, allowing us to loosen the over-restrictive i.i.d. assumption.

We explore the effect of three different prior probabilities in Equation 25, namely (i) the categorical distribution, by setting $p(\mathbf{c}; \boldsymbol{\pi}) = \prod_i Cat(c_i|\boldsymbol{\pi})$; (ii) the uniform distribution, by fixing $\pi_i = 1/K$ for $i = 1 \dots K$; and (iii) the multivariate noncentral hypergeometric distribution. We compare them on three different MNIST versions (LeCun & Cortes, 2010). The first version is the standard dataset which has a balanced class distribution. For the second and third dataset version we explore different ratios of subsampling for half of the classes. The subsampling rates are 80% in the moderate and 60% in the severe case. In Table 1 we evaluateevaluate the methods with respect to their clustering accuracy (Acc), normalized mutual information (NMI) and adjusted rand index (ARI).

As can be seen the hypergeometric prior distribution shows fairly good clustering performance in all datasets. It is also assuring that the model is able to learn the weights, $\boldsymbol{\pi}$, which reflect the subsampling rates of each cluster (see Figure 4). Although the uniform distribution performs reasonably good, it assumes the clusters have equal importance, hence it might fail in more complex scenarios. The categorical distribution, on the other hand, has subpar performance compared to the uniform distribution even in the moderate subsampling setting. This might be due to the additional complexity given by the learnable cluster weights, which results in unstable results. On the contrary,

the additional complexity does not seem to affect the performance of the proposed hypergeometric prior, but rather boost its clustering performance, especially in the imbalanced dataset.

### C.3.1 MODEL

We follow a deep variational clustering approach as described by Jiang et al. (2016). In particular, assuming the generative process described in Equations 25, 26 and 27, we can write the join probability of the data, also known as the likelihood function, as

$$p(X) = \sum_{\mathbf{c}} \int_{\mathbf{z}} p(X, Z, \mathbf{c}) = \sum_{\mathbf{c}} \int_{\mathbf{z}} p(\mathbf{c}; \boldsymbol{\pi}) p(X|Z) p(Z|\mathbf{c}) = \sum_{\mathbf{c}} p(\mathbf{c}; \boldsymbol{\pi}) \prod_i \int_{\mathbf{z}_i} p(\mathbf{x}_i|\mathbf{z}_i) p(\mathbf{z}_i|c_i) \tag{28}$$

Different from Jiang et al. (2016), the prior probability $p(\mathbf{c}; \boldsymbol{\pi})$ cannot be factorized as $p(c_i; \boldsymbol{\pi})$ for $i = 1, \ldots, K$ are not independent. By using a variational distribution $q_\phi(Z, \mathbf{c}|X)$, we have the following evidence lower bound

$$\log p(X) \geq E_{q_\phi(Z, c, |X)} \left[ \log \left( \frac{p(\mathbf{c}; \boldsymbol{\pi}) p(X|Z) p(Z|\mathbf{c})}{q_\phi(Z, \mathbf{c}, |X)} \right) \right] = \mathcal{L}_{ELBO}. \tag{29}$$

For sake of simplicity, we assume the following amortized mean-field variational distribution, as in previous work (Jiang et al., 2016; Dilokthanakul et al., 2016):

$$q_\phi(Z, \mathbf{c}|X) = q_\phi(Z|X) q_\phi(\mathbf{c}|X) = \prod_i q_\phi(\mathbf{z}_i|\mathbf{x}_i) q_\phi(c_i|\mathbf{x}_i). \tag{30}$$

From where it follows

$$\mathcal{L}_{ELBO} = E_{q_\phi(Z|X) q_\phi(\mathbf{c}|X)} \left[ \log p(\mathbf{c}|\boldsymbol{\pi}) + \log p(X|Z) + \log p(Z|\mathbf{c}) - \log q_\phi(Z, \mathbf{c}|X) \right] \tag{31}$$

$$= E_{q_\phi(Z|X) q_\phi(\mathbf{c}|X)} \left[ \log p(\mathbf{c}|\boldsymbol{\pi}) \right] + E_{q_\phi(Z|X)} \left[ \log p(X|Z) \right] + E_{q_\phi(Z|X) q_\phi(\mathbf{c}|X)} \left[ \log p(Z|\mathbf{c}) \right]$$

$$- E_{q_\phi(Z|X)} \left[ \log q_\phi(Z|X) \right] - E_{q_\phi(Z|X) q(\mathbf{c}|X)} \left[ \log q_\phi(\mathbf{c}|X) \right]. \tag{32}$$

In the ELBO formulation all terms, except the first one, can be efficiently calculated as in previous work (Jiang et al., 2016). For the remaining term, we rely on the following sampling scheme

$$E_{q_\phi(Z|X) q_\phi(\mathbf{c}|X)} \left[ \log p(\mathbf{c}|\boldsymbol{\pi}) \right] = \sum_{\mathbf{c}} \int_Z q_\phi(Z|X) q_\phi(\mathbf{c}|X) \log p(\mathbf{c}|\boldsymbol{\pi})) \tag{33}$$

$$= \sum_{\mathbf{c}} \prod_i \int_{\mathbf{z}_i} q_\phi(\mathbf{z}_i|\mathbf{x}_i) q_\phi(c_i|\mathbf{x}_i) \log p(\mathbf{c}|\boldsymbol{\pi}) \tag{34}$$

$$= \sum_{l=1}^{L} \log p(\mathbf{c}^l|\boldsymbol{\pi}), \tag{35}$$

where we use the SGVB estimator and the Gumbel-Softmax trick (Jang et al., 2016) to sample from the variational distributions $q_\phi(\mathbf{z}_i|\mathbf{x}_i)$ and $q_\phi(c_i|\mathbf{x}_i)$ respectively. The latter is set to a categorical distributions with weights given by:

$$p(c_i|\mathbf{z}_i; \boldsymbol{\pi}) = \frac{\mathcal{N}(\mathbf{z}_i|\boldsymbol{\mu}_{c_i}, \boldsymbol{\sigma}_{c_i}^2) \pi_{c_i}}{\sum_k \mathcal{N}(\mathbf{z}_i|\boldsymbol{\mu}_k . \boldsymbol{\sigma}_k^2) \pi_k}, \tag{36}$$

$L$ is the number of Monte Carlo samples and it is set to 1 in all experiments.

### C.3.2 IMPLEMENTATION DETAILS

To implement our model we adopted a feed-forward architecture for both the encoder and decoder of the VAE with four layers of $500, 500, 2000, D$ units respectively, where $D = 10$. The VAE is pretrained using the same layer-wise pretraining procedure used by Jang et al. (2016). Each data set is divided into training and test sets, and all the reported results are computed on the latter. We employed the same hyper-parameters for all experiments. In particular, the learning rate is set to 0.001, the batch size is set to 128 and the models are trained for 1000 epochs. Additionally, we used an annealing schedule for the temperature of the Gumbel-Softmax trick.

