# OpenReview forum: "Continuous Relaxation For The Multivariate Noncentral Hypergeometric Distribution"
_ICLR.cc/2022/Workshop/OSC — ICLR2022 OSC  Poster_

### Official Review · Reviewer_quts · 2022-03-05
**Extending work on relaxing discrete distributions**

**Rating:** 2
**Confidence:** 2

**Review:**

This work proposes a relaxation of the (multivariate) non-central hypergeometric distribution, which can be used for modeling the number of successes when elements from more than one class can be drawn. The relaxation is based on the Gumbel-Softmax trick. Experiments in this paper on weakly-supervised learning demonstrates that it works well in practice.

The paper is mostly well written. The contribution is clear, namely a relaxation of the non-central hypergeometric distribution. While a lot of its parts have been used before in other relaxation methods (e.g. sequential application of Gumbel-Softmax's), it is still not a trivial extension to the specific distribution presented in the paper, and probably of interest to a part of the audience. The description of the method in Section 2 is detailed enough for comprehending all the steps and parts necessary for implementing this approach.

In its current form, the paper has three aspects that can be improved. Firstly, the introduction should give clearer examples on where this distribution is useful, especially in a machine learning setting. While the second paragraph of the introduction gives broad examples of biology to social sciences, none of them are very specific, and the experiments conducted in the paper focus on a different aspect. It would be better to align this part with the experiments. The second aspect that can be improved is the clarity of the experiments. For a lot of the experimental setup, the reader is pointed to different papers, and the overall goal/task of the models is not clear to readers who are not familiar with the works. It is advised to clarify the experimental setup (which also combines with the first aspect in the introduction) to open up the paper to a wider audience. Finally, the discussion on the accuracy of the learned representations is not completely fair. In the plot in Figure 1b, the proposed method (HypergeometricVAE) is consistently outperformed in all 4 settings by both baselines, where the LabelVAE's lower std is always above the HypergeometricVAE's upper std performance. The difference is in some cases more than 15%. The paper does not acknowledge or discuss that, but instead says that all methods perform comparatively, which is not fully fair. It is not bad to have worse results than baselines here, but there should be a fair discussion of the numbers, and why the proposed method is lacking behind the two baselines.

Overall, the contributions of this paper outweigh the drawbacks, assuming that the authors will tackle the aspects mentioned above for the final paper version.

Minor points:
* Page 1, Introduction, line 4: last citation missing brackets.
* Equation 1: the class weights omega_l/_r have not been introduced at this stage.

---

### Official Review · Reviewer_UrgT · 2022-03-10
**Sensible sampling strategy, non-intuitive experiments**

**Rating:** 2
**Confidence:** 3

**Review:**

The paper discusses the relaxation the multivariate non-central hypergeometric distribution. This is the distribution whereby one has $N$ objects in $c$ classes, with $m_i$ objects of class $i$. From these, one then samples $n$ objects without replacement, where objects in class $i$ are sampled with weight $\omega_i$. The authors reformulate this multivariate case as a sequence of $c$ bivariate cases. In step $i$, they assume the classes $\lt i$ have been sampled, then condition on that result and sample the remaining from either class $i$ or the classes $\gt i$. Each bivariate case is then a categorical over the number of samples in class $i$, which is relaxed via the Gumbel Soft-Max trick.

For the experiments, the paper considers the task of observing pairs of images, generated from a $d$ dimensional latent, whereby the latents differ in $k$ dimensions (sampled independently) and are equal on $d-k$ dimensions. The goal is to learn a VAE with an encoder that maps to a latent space that differs from the ground truth only via a permutation and element-wise diffeomorphism. For each pair, $k$ differs and is unknown, thus must be inferred by the encoder. The authors apply their method, in the bivariate case, with $d$ samples from $2d$ objects, with $d$ in class "latents equal" and $d$ in class "latents differ". The weights of the classes is a learned function of the encoder. Then a number $k$ is sampled in the class "latents differ", which are then heuristically matched to $k$ of the $d$ dimensions.

The relaxation strategy is makes sense to me. However, I find the application in the experiment very non-intuitive. Firstly, they only experiment with the bivariate case, while they propose a multi-class method. Secondly, why is this a hypergeometric problem in the first place? Why first predict a number $k$ and then with an unelegant heuristic map this to $k$ of the $d$ dimensions? A much more natural representation seems to me to just treat it as $d$ binary variables: whether the dimension is equal or not. The probabilities could then e.g. be computed learnably from the $\gamma$ vector. This would be simpler and avoid the heuristic of post-hoc selection of the $k$ dimensions. In any case, I'm sure that one can find experiments in which the probabilistic problem is clearly a multivariate non-central hypergeometric distribution, which I'd recommend to evaluate the method on.

Furthermore, there are some serious presentation issues. The multivariate non-central hypergeometric distribution itself is not defined in the main paper, only in the appendix. The $\omega$ variables in equation 1 are not defined in the main paper. The variables $\alpha_i$ are used in equation 2, but not defined. I'd recommend a big restructuring, in which first the mutivariate case is explained (including the PMF in the main paper), then the sequential sampling (including sec 2.3) and only then the relaxation.

I weakly recommend acceptance of this paper because the general method seems sensible, but I suggest that the authors evaluate their method on more fitting problems and clarify the presentation for the final version.

---

### Decision · Program_Chairs · 2022-03-23

Accept (Poster)